# OpenReview forum: "CBV: Clean-label Backdoor Attacks on Vision Language Models via Diffusion Models"
_ICML.cc/2026/Conference — ICML 2026 regular_

### Official Review · Reviewer_1PJC · 2026-02-20

**Soundness:** 4
**Presentation:** 3
**Significance:** 3
**Originality:** 4
**Overall Recommendation:** 5
**Confidence:** 5

**Summary:**

The authors investigate the vulnerability of Vision-Language Models (VLMs) to backdoor attacks and propose CBV, a clean-label attack framework based on diffusion models. The authors generate natural poisoned samples via score-guided diffusion while leveraging multimodal guidance and GradCAM-guided masks to improve effectiveness and stealthiness. Experiments on MSCOCO and VQA v2 demonstrate high attack success rates with minimal impact on normal functionality.

**Compliance With Llm Reviewing Policy:**

Affirmed.

**Final Justification:**

The rebuttal successfully resolved my main concerns. I lean towards accepting this paper.

**Key Questions For Authors:**

Overall, the authors propose a novel backdoor attack that leverages diffusion models to generate clean-label poisoned samples. The paper also provides theoretical analysis and includes extensive experiments in both the main text and the appendix. My main concerns are as follows:
1. Does the model scale affect the attack performance? Is the attack still effective on larger models, such as 32B?
2. Please specify the experimental hardware setup to facilitate reproducibility.

**Limitations:**

Yes

**Strengths And Weaknesses:**

### Strengths
1. The idea of using diffusion models to generate poisoned samples is novel. Compared with prior methods that directly optimize images, leveraging diffusion models improves both attack effectiveness and stealthiness.
2. The paper is clearly written and allows readers to easily understand the core idea, namely injecting poisoned information during the reverse denoising process.
3. The experiments are comprehensive, providing thorough comparisons with various existing backdoor attacks on VLMs, while also considering time overhead and defense methods.

### Weaknesses
1. The authors mainly consider models at the 7B scale, while larger-parameter models are not evaluated; therefore, the impact of model size on the attack remains unclear.
2. The authors do not specify the exact hardware configuration (e.g., GPU) used in the experiments, which makes the evaluation of computational overhead less transparent.

---

> ### Author Rebuttal · Authors · 2026-03-26
>
> Thank you for your recognition of our paper and your valuable suggestions. Below are our responses.
>
> ### 1. Impact of model scale on attack performance
> To address whether model scale affects attack performance, we further conduct poisoning experiments on larger VLMs, including InternVL3-8B, InternVL3-14B, and Qwen2.5-VL-32B.
>
> As shown in Table 1, CBV maintains consistently high ASR across these large-scale models. This indicates that our method generalizes well to larger models, and increasing model size does not significantly affect attack effectiveness.
>
> **Table 1. Attack performance across different model scales**
>
> |Model|Cap ASR（%）|VQA  ASR（%）|
> |---|---|---|
> |InternVL3-8B|85.00|81.96|
> |InternVL3-14B|86.43|82.28|
> |Qwen2.5-VL-32B|83.71|82.20|
>
> ### 2. Experimental hardware
> We apologize for the missing hardware details. In our experiments involving diffusion-based poisoned image generation, we use an NVIDIA RTX 4090 (24GB) GPU for all computations.We will include this information in the revised version in attack configuration.

---

> > ### Author Rebuttal · Reviewer_1PJC · 2026-04-03
> >
> > Thanks for the additional experiments and clarifications. The authors have added experiments on larger VLMs with more parameters, discussed the impact of model scale on the attack performance, and provided detailed hardware configurations, which further improve the reproducibility of the experiments. My concerns have been addressed.

---

### Official Review · Reviewer_ntM9 · 2026-02-26

**Soundness:** 3
**Presentation:** 4
**Significance:** 3
**Originality:** 4
**Overall Recommendation:** 5
**Confidence:** 5

**Summary:**

The authors propose a novel clean-label backdoor attack for VLMs that leverages score matching in diffusion models to produce stealthy poisoned samples. By integrating textual guidance and a GradCAM-guided mask, the authors constrain perturbations to semantically important regions, improving both realism and attack reliability. The authors validate their approach on MSCOCO and VQA v2, demonstrating effective attacks without degrading model utility.

**Compliance With Llm Reviewing Policy:**

Affirmed.

**Final Justification:**

All my concerns have been well addressed, and I strongly recommend acceptance.

**Key Questions For Authors:**

Overall, this paper proposes an effective and stealthy clean-label backdoor attack on VLMs, supported by comprehensive experiments. The main concerns are: (1) whether the diffusion model used to generate poisoned samples needs to be retrained, or if a normally pretrained diffusion model can be directly used; and (2) whether the attack target influences the attack performance. I would consider revising my score if the authors address these issues.

**Limitations:**

Yes

**Strengths And Weaknesses:**

Strengths
1.This paper presents the first effective clean-label backdoor attack on VLMs, achieving ASR comparable to dirty-label attacks. The overall performance is impressive.
2.The idea of using diffusion-based score matching to generate poisoned samplehs is novel and well motivated. In addition, the GradCAM-guided masking further enhances the stealthiness of the attack.
3.The experimental evaluation is comprehensive. The method is tested across multiple VLMs and tasks, and extensive ablation studies are provided in both the main experiments and the supplementary materials.

Weaknesses
1.The attack mainly targets the human-to-dog transformation. Does the choice of attack target influence the attack performance?
2.Does the diffusion model used in this work need to be retrained, or can it directly load standard pretrained diffusion model weights? This is also important for evaluating the computational overhead

---

> ### Author Rebuttal · Authors · 2026-03-26
>
> Thank you for your kind recognition of our paper and your valuable comments. Below are our responses to your questions.
>
> ### 1. Diffusion model retraining
> Our method directly uses a standard pre-trained diffusion model without any retraining or fine-tuning. We modify the score function during the reverse diffusion process rather than altering the model parameters. Specifically, during generation, we incorporate both the visual features of the trigger image and its corresponding textual semantics as multimodal guidance to steer the denoising score function, thereby generating poisoned samples. This process does not change the model parameters, but only the generation objective.
>
> ### 2. Impact of attack targets
> We have discussed the impact of attack targets in the appendix (see Appendix D, Table 11). Specifically, we evaluate 8 different target substitutions, all achieving over 80% ASR. This demonstrates that our method is not sensitive to the choice of target and generalizes well across different targets.

---

> > ### Author Rebuttal · Reviewer_ntM9 · 2026-04-01
> >
> > Thanks for the response. My concerns have been fully addressed. Overall, the idea of the paper is intuitive and the experiments are comprehensive. I have no further comments.

---

> > > ### Author Response · Authors · 2026-04-03
> > >
> > > Thank you very much for your valuable comments and professional review of our work.

---

### Official Review · Reviewer_vCbx · 2026-03-03

**Soundness:** 4
**Presentation:** 4
**Significance:** 3
**Originality:** 3
**Overall Recommendation:** 5
**Confidence:** 5

**Summary:**

The authors present CBV, a diffusion-based clean-label backdoor attack on VLM attacks. The authors modify the reverse diffusion process to inject trigger features and introduce multimodal guidance and GradCAM-guided masking to preserve visual naturalness. Extensive evaluations across four VLMs show that the method achieves strong ASR while maintaining standard task performance.

**Compliance With Llm Reviewing Policy:**

Affirmed.

**Ethical Review Concerns:**

No ethical concerns.

**Final Justification:**

See the Acknowledgement.

**Key Questions For Authors:**

1. What is the threshold used in GM, and why is this threshold chosen instead of simply selecting the top (or bottom) percentage of regions?
2. Does using different diffusion models to generate poisoned samples influence the attack effectiveness?

**Limitations:**

yes

**Strengths And Weaknesses:**

Strengths
* The authors propose a novel clean-label backdoor attack targeting VLMs. Previous clean-label backdoor attacks mainly focused on image classification models, and this work fills the gap for VLMs.
* The threat model is well-established and realistic, with a reasonable assumption about the attacker’s capabilities.
* The experimental results are impressive. Compared with prior clean-label backdoor attacks, the proposed method achieves significant improvements across multiple VLMs and datasets.

Weaknesses
* The paper uses GM to restrict modifications to a limited region of the image. However, based on the results in Fig. 5, it seems that the authors do not simply select the top percentage of regions, but rather modify the middle-importance regions according to Grad-CAM. The specific threshold or selection criterion is not clearly described in the paper.
* The method mainly uses SD1.4 as the diffusion model for generation, but the impact of using other diffusion models is not discussed. A key question is whether the choice of diffusion model affects the attack performance.

---

> ### Author Rebuttal · Authors · 2026-03-26
>
> Thank you for your recognition of our paper and your valuable comments. Below are our responses.
>
> ### 1. Threshold range in GM
> Following prior work on Grad-CAM-based masks, we adopt normalized Grad-CAM values within the range of **0.3–0.7** to construct the mask. This choice aims to balance semantic coverage and perturbation locality.Specifically, a high threshold tends to focus on a few highly salient regions, which highlights key discriminative areas but limits the space available for perturbation injection, making it difficult to effectively embed trigger features. In contrast, a low threshold includes more background or semantically irrelevant regions, which may introduce visual artifacts and harm the naturalness and stealthiness of poisoned samples.The 0.3–0.7 range provides a better trade-off by sufficiently covering semantically relevant regions, enabling stable trigger injection while maintaining both attack effectiveness and visual stealthiness.
>
> ### 2. Impact of different diffusion models on CBV
> We evaluate three mainstream pre-trained diffusion models in Appendix H, including Stable Diffusion v1.4, Stable Diffusion v2.1, and SDXL v1.0 (see Tables 12 and 13).
> Results show that CBV consistently achieves over 80% ASR across all diffusion models. This indicates that the effectiveness of CBV is not sensitive to the choice of diffusion model and can generalize well across different pre-trained generators.

---

> > ### Author Rebuttal · Reviewer_vCbx · 2026-04-01
> >
> > Thank you for the authors’ detailed response to our concerns. The authors have provided further clarification on the choice of the Grad-CAM threshold range and the impact of different diffusion models on CBV, supported by additional experimental results in the appendix. My concerns have been addressed, and I have decided to increase my score.

---

> > > ### Author Response · Authors · 2026-04-03
> > >
> > > Thank you for considering improving your score. We greatly appreciate your constructive feedback and the time you have invested in reviewing our work. If there are any further questions or suggestions, we would be happy to address them and revise the manuscript accordingly.

---

### Official Review · Reviewer_X2UT · 2026-03-10

**Soundness:** 3
**Presentation:** 3
**Significance:** 3
**Originality:** 3
**Overall Recommendation:** 4
**Confidence:** 3

**Summary:**

This paper investigates the vulnerability of Vision-Language Models (VLMs) to clean-label backdoor attacks. The authors propose a novel framework, CBV, which utilizes Diffusion Models to generate poisoned samples. Unlike traditional "dirty-label" attacks that require mismatched image-text pairs, CBV embeds backdoor triggers (Universal Adversarial Perturbations, UAP) into images through a score-matching process guided by GradCAM masks. This ensures that the poisoned images remain visually natural and semantically aligned with their original labels, making the attack difficult to detect by both human inspection and automated defense mechanisms.

**Compliance With Llm Reviewing Policy:**

Affirmed.

**Final Justification:**

1. This paper proposes a clean-label backdoor attack against VLM without altering text labels, addressing a major limitation of prior VLM security research.
2. This paper leverages the score-matching property of Diffusion Models to generate backdoor-poisoned images, which is a creative and technically sound approach.

Weaknesses have been addressed by the rebuttal.

**Key Questions For Authors:**

- While the paper evaluates existing defenses, it would be interesting to see a discussion on potential new defense strategies specifically designed to detect the backdoor samples. In addition, how robust is this method to model pruning attacks?

- A critical question arises regarding the inherent efficacy of Universal Adversarial Perturbations (UAP). Specifically, the authors should clarify the performance of UAP-generated poisoned samples when applied to a "clean" VLM that has not been embedded with a backdoor. In a standard adversarial attack setting, what is the baseline Attack Success Rate (ASR) of these UAP samples?

**Limitations:**

No

- The authors should discuss the trade-off between the poisoning ratio, the stealthiness of the generated images, and the ultimate attack effectiveness.

- For a backdoor attack method, the authors should discuss some potential defense methods to mitigate the vulnerability of the VLLM model.

**Strengths And Weaknesses:**

## Strengths:

1. This paper proposes a clean-label backdoor attack against VLM without altering text labels, addressing a major limitation of prior VLM security research.

2. This paper leverages the score-matching property of Diffusion Models to generate backdoor-poisoned images, which is a creative and technically sound approach.

3. The authors evaluate the attack on multiple VLMs across different datasets (MSCOCO, VQA v2). The solid experiments demonstrate the effectiveness and robustness to common attacks, e.g. , finetuning.

4. The paper demonstrates that poisoning samples using a surrogate CLIP model effectively transfers to diverse downstream VLMs like LLaVA, MiniGPT-v2, and Qwen-VL.

## Weaknesses:

1. Regarding the model utility, the experimental results in Table 2 and Table 3 reveal that the proposed UAP-based poisoning image generation method leads to a noticeable performance degradation compared to the clean baseline models.

2. Another major concern regarding the feasibility of the proposed method lies in the high poisoning ratio required for a successful attack. In the experiments, the authors utilize a poisoning ratio of 10%, which may not be a realistic or achievable goal in actual large-scale VLM training scenarios.

3. A major concern regarding the perceptual stealthiness of the proposed attack. While the authors claim that the diffusion-based generation produces natural-looking poisoned examples, the quantitative results in Table 10 suggest otherwise. Specifically, the PSNR values are below 25 dB, which typically indicates perceptible distortion and artifacts in the image processing community.

---

> ### Author Rebuttal · Authors · 2026-03-26
>
> Thank you for your comments on our paper. We believe some points may benefit from further clarification, as detailed below.
> ### 1. Model utility
> There may be a misunderstanding regarding the performance degradation caused by backdoor models. In fact, it is unavoidable that the normal functionality of backdoor models is slightly lower than that of clean models. As shown in Table 2 and Table 3, prior methods (e.g., TrojVLM [1], BadToken [2]) also introduce performance degradation. For example, the average VQA accuracy of TrojVLM is 74.74%, while our method achieves a VQA accuracy of 76.52%. Moreover, other reviewers have also acknowledged that our method preserves model utility, noting that it “achieves strong ASR while maintaining standard task performance” (Reviewer vCbx), “effective attacks without degrading model utility” (Reviewer ntM9), and “high attack success rates with minimal impact on normal functionality” (Reviewer 1PJC).
>
> ### 2. Poisoning ratio
> We would like to clarify that the poisoning ratio used in our paper is 5%, rather than the 10% mentioned in your comment (see Page 6, Line 300, Attack Configuration). We also evaluate the attack performance across a range of poisoning ratios (0%–10%) in Fig. 6. Even at a relatively low ratio, our method remains effective. More broadly, a 5% poisoning ratio is a standard setting in prior work. For example, TrojVLM adopts 5% as its default, while some methods (e.g., CBAVR [3] need 30%) require substantially higher ratios.
>
> ### 3. Stealthiness
> We clarify that our stealthiness strategy is achieved by generating natural-looking poisoned examples, rather than enforcing strict pixel-level similarity to the original images.In our backdoor attack scenario, users directly download poisoned datasets from websites. These datasets consist of a mixture of poisoned and clean images. The poisoned images are generated based on some original clean images; however, the corresponding clean versions of these poisoned images are not included in the poisoned dataset. Therefore, ensuring that the images are natural-looking is sufficient to achieve stealthiness (as also discussed in Color Backdoor [4]).
> In Table 10, we report PSNR to evaluate the impact of Grad-CAM generated by different surrogate models on the mask, rather than to measure stealthiness based on the similarity between triggered images and the original images.
> As shown in Figs. 5 and 9, the poisoned images are nearly indistinguishable from clean ones to the human eye. Moreover, other reviewers have also recognized the stealthiness of our method, describing it as having “visual naturalness” (Reviewer vCbx) and demonstrating both “effectiveness and stealthiness” (Reviewer 1PJC).
>
> ### 4. Robustness to pruning
> We have already evaluated the robustness under pruning followed by fine-tuning (fine-pruning) in Fig. 8(b). To further investigate robustness specifically against model pruning defense, we additionally include RNP [5] experiments on the backdoor Qwen-2.5-VL-7B. As shown in Table 1, even with increasing pruning ratios, the ASR remains consistently high, while the drop in VQA accuracy is moderate, indicating that pruning is ineffective in removing the backdoor.
>
> **Table 1. Pruning Robustness (Qwen-2.5-VL-7B)**
> |Pruning Ratio (%) |ASR (%) |VQA (%) |
> |---|---|---|
> |0|83.58|79.36|
> |2|83.21|78.42|
> |5|82.67|77.13|
> |8|82.64|76.02|
> |10|82.32|75.08|
>
> ### 5. Inherent efficacy of UAP
> In our method, UAP is not used to directly induce targeted outputs, but to enhance the model’s ability to learn discriminative trigger features during training, thereby facilitating the backdoor. To verify this, we evaluate UAP on clean VLMs under the same settings. As shown in Table 2, the ASR remains below 5% across all models, indicating that UAP alone cannot induce targeted outputs. This confirms that the attack success mainly stems from the injected backdoor, while UAP primarily serves to strengthen trigger learning rather than directly perform the attack.
>
> **Table 2. UAP on Clean VLMs**
> |Model|ASR (%) |
> |---|---|
> |LLaVA-7B|3.12|
> |MiniGPT|2.87|
> |BLIP-7B|4.03|
> |Qwen-7B|3.45|
>
> ### 6. Potential defense
>  Thank you for your suggestion. The primary contribution of this paper is to propose a novel clean-label backdoor attack for VLMs, rather than developing defense methods. Exploring more effective defenses is an important direction for our future work.
>
> [1] Lyu, Weimin, et al. "Trojvlm: Backdoor attack against vision language models." Proceedings of ECCV. 2024.
>
> [2] Yuan, Zenghui, et al. "Badtoken: Token-level backdoor attacks to multi-modal large language models." Proceedings of CVPR. 2025.
>
> [3] Zhao, Shihao, et al. "Clean-label backdoor attacks on video recognition models." Proceedings of CVPR. 2020.
>
> [4] Jiang, Wenbo, et al. "Color backdoor: A robust poisoning attack in color space." Proceedings of CVPR. 2023.
>
> [5] Li, Yige, et al. "Reconstructive neuron pruning for backdoor defense." Proceedings of ICML. 2023.

---

> > ### Author Rebuttal · Reviewer_X2UT · 2026-04-03
> >
> > Thank you for the detailed reply, which clarified my doubts, and I will improve my final score.

---

> > > ### Author Response · Authors · 2026-04-03
> > >
> > > Thank you for your thoughtful reconsideration of our paper and for considering improving your score. We are pleased that our response has resolved your questions. We greatly appreciate your constructive feedback and the time you have invested in reviewing our work.

---

### Decision · Program_Chairs · 2026-04-30

**Decision:**

Accept (regular)

**Comment:**

The paper proposes a clean-label poisoning attack against vision-language models by cleverly exploiting the scoring mechanism of a diffusion model. The paper is well-written, the idea is sufficiently novel, and the experimental results are convincing. All the reviewer concerns were addressed during the rebuttal and the reviewers have acknowledged the same. Thus, the paper is a solid contribution to ICML.